ᵃ | **Open Peer Review** | Bacteriology | Research Article

# Differential vasoproliferative traits of *Bartonella henselae* strains associated with autotransporter BafA variants

Yuka Kondo,[1,2] Masahiro Suzuki,[2] Shingo Sato,[3] Soichi Maruyama,[3] Akiko Sei,[2] Xingyan Ma,[1] Kota Nakano,[2] Yohei Doi,[2,4,5] Kentaro Tsukamoto[1]

**ABSTRACT** *Bartonella henselae,* a Gram-negative facultative intracellular bacterium, is the etiological agent of cat-scratch disease and also causes bacillary angiomatosis in immunocompromised individuals. Although the ability to promote vascular endothelial cell proliferation differs among *Bartonella* species, variations among strains within *B. henselae* remain unclear. *Bartonella* angiogenic factor A (BafA) and *Bartonella* adhesin A (BadA) have been identified as autotransporters of *B. henselae* that are involved in endothelial cell proliferation. Although strain-specific differences in the expression of BadA and the VirB/D4 type IV secretion system have been reported, BafA expression among *B. henselae* strains has yet to be examined. Therefore, the present study investigated the proliferation-promoting ability of 13 *B. henselae* strains from several sources in human umbilical vein endothelial cells (HUVECs). We identified BafA variants 1 and 2 based on the deduced amino acid sequences of its passenger domain. The recombinant proteins of both variants exhibited similar proliferation activity against HUVECs. However, BafA variant 2 strains showed cytotoxicity at a high bacterial inoculum in a direct coculture with HUVECs, which was attenuated in an indirect coculture. These strains, in contrast to BafA variant 1 strains, highly expressed BadA and exhibited bacterial aggregation. Based on a core genome SNP analysis of 50 *B. henselae* strains, the BafA variant types corresponded to clades 1–4. These results indicate that vasoproliferative traits differ among *B. henselae* clades based on the variant types. Therefore, this study provides a new conceptual framework in which the clades of *B. henselae* may predict their pathogenicity in humans.

**IMPORTANCE** *Bartonella* species including *Bartonella henselae*, *Bartonella quintana*, and *Bartonella bacilliformis* cause vasoproliferative lesions. Their proliferation-promoting ability in vascular endothelial cells differs among *Bartonella* species; however, it is unclear whether these differences exist among *B. henselae* strains. We herein showed that *B. henselae* strains exhibited variable proliferation-promoting ability and cytotoxicity in vascular endothelial cells, which corresponded to the *bafA* gene variants possessed by the strains. The expression levels of *Bartonella* angiogenic factor A (BafA) and *Bartonella* adhesin A, as well as the degree of proliferation-promoting ability and cytotoxicity in endothelial cells, varied among the strains. A core genome SNP analysis of strains using whole genome sequencing data divided *B. henselae* strains into four clades, with each clade corresponding to BafA variants 1–4. These results suggest the differential vasoproliferative potency of *B. henselae*, with potential implications in clinical management, including risk stratification and predictions of the clinical course.

**KEYWORDS** *Bartonella*, autotransporter proteins, angiogenesis, endothelial cells, cell proliferation

Address correspondence to Kentaro Tsukamoto, tsukamoto@biken.osaka-u.ac.jp.

The authors declare no conflict of interest.

See the funding table on p. 15.

The genus *Bartonella* is Gram-negative facultative intracellular parasitic bacteria. Thirty-nine species and three subspecies of *Bartonella* have been identified so far, and at least 13 species are known to cause zoonotic infections (1, 2). Among them, *Bartonella henselae* is the most common species in human bartonellosis and is typically acquired through direct contact with its natural reservoir, cats, or less commonly via cat flea infestation (3–5). In immunocompetent individuals, it manifests as cat-scratch disease, a self-limiting condition (6). However, in immunocompromised individuals infected with *B. henselae*, the bacterium may cause bacillary angiomatosis through the induction of angiogenesis (7, 8). *B. henselae* induces hemangioma-like lesions by inducing proangiogenic cytokines from immune cells and exerting anti-apoptotic effects on host vascular endothelial cells, both of which contribute to endothelial cell proliferation (9–13). Therefore, *B. henselae* likely sustains infection and exhibits its pathogenicity by promoting the proliferation of host vascular endothelial cells (14). A trimeric autotransporter adhesin (TAA) of *B. henselae*, *Bartonella* adhesin A (BadA), has been shown to induce proangiogenic responses and also contribute to host cell adhesion, bacterial auto-aggregation, and biofilm formation through its adhesive properties (15–17). We previously identified another virulence autotransporter protein, *Bartonella* angiogenic factor A (BafA), produced by *B. henselae* (18).

In our previous studies, we demonstrated the differential effects of BafA on vascular endothelial cells among *Bartonella* species (19–21); however, systematic comparisons between different strains within *B. henselae*, the cardinal *Bartonella* species associated with human infection, have yet to be conducted. Chang et al. reported that the interaction between *B. henselae* and endothelial cells differed depending on the strains, and only the mitochondrial apoptotic pathway was investigated as the potential cause of this finding (22). *B. henselae* has been isolated from humans and cats in countries and regions worldwide (23), and more than 400 strains have been registered in the PubMLST database to date. These strains have been classified into 38 sequence types (ST) by multilocus sequencing typing (MLST; https://pubmlst.org/organisms/bartonella-henselae). Some of these STs have been associated with specific host animals (24). A Spanish study showed that ST5, ST6, and ST9 strains were associated with cats, whereas ST1, ST5, and ST8 strains were mostly isolated from human cases of bartonellosis (25). Similarly, a UK study reported that ST6 strains were most commonly isolated from cats, whereas ST2, ST5, and ST8 strains were more commonly isolated from humans than from cats (26). However, the genotypic differences that define these characteristics have not yet been elucidated. Furthermore, the potential implications of these differences in the pathogenicity and infectivity of *B. henselae* remain unclear.

Therefore, the present study investigated whether differences exist in the proliferative ability against vascular endothelial cells among *B. henselae* strains and, if so, examined the mechanisms responsible for these differences. The results obtained showed that the proliferation-promoting effect on vascular endothelial cells, as well as the degree of BafA secretion and BadA expression, differed among the strains tested, and they, in turn, were associated with the specific BafA variants produced. Therefore, the diversity of proliferation-promoting effects on vascular endothelial cells observed among *B. henselae* strains may be attributed to differences in their phylogenetic clades. The present study provides a new conceptual framework in which the clades of *B. henselae* may predict their pathogenicity in humans.

## RESULTS

### Induction of cell proliferation varies among strains of *B. henselae*

To examine the proliferation ability of *B. henselae* strains, we first infected human umbilical vein endothelial cells (HUVECs) with each strain at multiplicities of infection (MOIs) of 100, 300, and 1,000 and evaluated the proliferation of HUVECs (Fig. 1). At a MOI of 100, HUVEC proliferation was significantly greater with strains Oki.cat48, i6-1, HJ53, and PL18 than with the medium control. When the MOI was increased to 300, two additional strains, Houston-1 and 87–66, also significantly enhanced cell proliferation. At

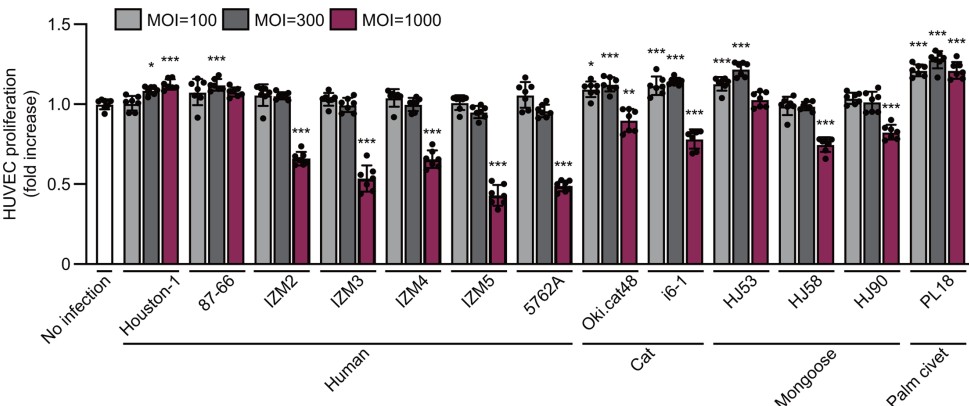

**FIG 1** Endothelial cell proliferation in response to *B. henselae* infection. HUVECs were cocultured with various *B. henselae* strains (Houston-1, 87–66, IZM2-5, 5762A, Oki.cat48, i6-1, HJ53, HJ58, HJ90, and PL18) at MOIs of 100, 300, and 1,000 for 2 days. The results show relative fold changes in the cell count from uninfected controls. Bars represent means ± SDs (*n* = 7; circles indicate individual replicates). The significance of differences was examined using a one-way analysis of variance (ANOVA) with Dunnett's test: *, *P* < 0.05; **, *P* < 0.01; ***, *P* < 0.001.

this higher MOI, between 1.09-fold and 1.28-fold increases in HUVEC proliferation were observed across these cell proliferation-promoting strains. At a MOI of 1,000, Houston-1 and PL18 continued to show significant cell proliferative ability, whereas 87–66 and HJ53 no longer exhibited significant increases. In contrast, at this MOI, HUVEC proliferation was significantly impaired by strains IZM2, IZM3, IZM4, IZM5, 5762A, Oki.cat48, i6-1, HJ58, and HJ90, resulting in decreased cell numbers. These results indicate the different abilities of *B. henselae* strains to promote the proliferation of HUVECs at various MOIs and exhibit cytotoxicity at higher MOIs.

Previous studies reported that *B. henselae* infection induced the expression of proangiogenic cytokines in host cells (15, 27–30). To investigate potential factors responsible for the differences in cell proliferative capacity and cytotoxicity observed during HUVEC infection, we examined the mRNA expression levels of four cytokines in HUVECs infected with the representative strains. We selected two cell proliferative strains (Houston-1 and PL18) and two cytotoxic strains (IZM5 and 5762A). The cytokines examined were as follows: vascular endothelial growth factor (VEGF) and interleukin-8 (IL-8) as proangiogenic cytokines, and tumor necrosis factor-α (TNF-α) and IL-6 as proinflammatory cytokines. The results obtained showed that IL-8 expression levels significantly increased in strains PL18, IZM5, and 5762A, with Houston-1 showing a slight increase. However, no marked increases were observed in VEGF, TNF-α, or IL-6 (Fig. S1). None of the four cytokines exhibited differential expression patterns that correlated with the cell proliferation-promoting or cytotoxic properties of the strains.

## Recombinant BafA1 and BafA2 exhibit cell proliferation activity

To investigate whether differences in the cell proliferation-promoting ability of each strain were due to BafA, the presence or absence of the *bafA* gene was examined. An approximately 3.0 kb region encompassing the *bafA* gene, including the putative promoter region, was amplified by PCR. All 13 *B. henselae* strains possessed the *bafA* gene (Fig. 2A). BafA is an autotransporter protein with a total length of 874 amino acids, in which the passenger domain at the N terminus is secreted outside the bacterial cell and exhibits activity. Therefore, *bafA* amplicons were sequenced, and the deduced amino acid sequences of the passenger domains encoded by these genes were compared. Amino acid sequences were classified into BafA variant 1 (BafA1) and variant 2 (BafA2). Houston-1, 87–66, HJ53, and PL18 possessed BafA1, whereas IZM2, IZM3, IZM4, IZM5, 5762A, Oki.cat48, i6-1, HJ58, and HJ90 possessed BafA2. The amino acid sequences of strains within each BafA variant were identical; however, sequence similarity between

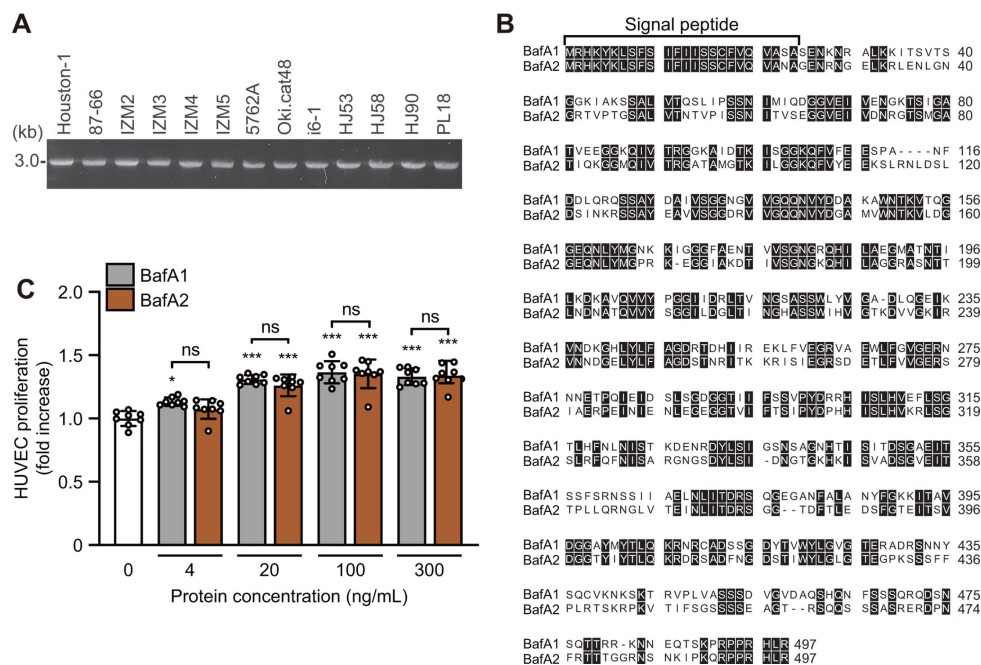

**FIG 2** BafA gene detection and variant comparison. (A) PCR detection of the *bafA* gene, including the putative promoter, in *B. henselae* strains. (B) Amino acid sequence alignment of passenger domains from BafA1 and BafA2. Conserved residues are shown in white on black. (C) HUVEC proliferation in response to recombinant BafA variants. Cells were treated with increasing doses for 2 days. The results show relative fold changes from untreated controls. Bars: means ± SDs (*n* = 8; open circles). A one-way ANOVA with Dunnett's test: ns, not significant; *, *P* < 0.05; ***, *P* < 0.001. Notations above bars indicate comparisons to the control group.

BafA1 and BafA2 was as low as 53% (Fig. 2B). To clarify whether the cell proliferation activities of BafA1 and BafA2 differed, we prepared recombinant BafA variants and added these recombinants to the culture medium of HUVECs in order to evaluate cell proliferation. The BafA1 and BafA2 recombinants both exhibited proliferation activity against HUVECs in a concentration-dependent manner (Fig. 2C). The proliferation activities of recombinant BafA1 and BafA2 against HUVECs did not significantly differ at each concentration tested. To verify that the observed proliferation of HUVECs was specifically induced by recombinant BafA and not by potential contaminants, we conducted control experiments (Fig. S2). HUVECs were treated with a recombinant BadA stalk, which did not affect cell growth. This result supports the specificity of BafA-induced proliferation. Furthermore, we compared the effects of the BafA treatment with those of VEGF, a known potent inducer of endothelial cell proliferation. The VEGF treatment resulted in a similar level of cell proliferation to that observed with the BafA treatment, underscoring the significant proliferative capacity of BafA. These results demonstrated that although BafA1 and BafA2 have distinct sequences, they are functionally equivalent and exhibit similar proliferation activities against HUVECs.

## Detection of BafA in cell lysates and culture supernatants

Based on these experimental results, we hypothesized that the expression and/or amount of BafA secreted in the culture supernatant may differ among the strains carrying the two variants. To test this hypothesis, we initially examined BafA expression in bacterial cells infecting HUVECs by western blotting (Fig. 3A). The results obtained revealed bands around 80–90 kDa, corresponding to intact BafA (including the passenger, linker, and translocator domains of autotransporters), and bands around 50 kDa, representing the passenger domain alone. These bands were detected at similar levels across all BafA1 strains. In contrast, BafA2 strains exhibited varying levels of the

80–90 kDa bands among strains, whereas the 50 kDa band was not detected in any of these strains. To investigate whether differences in BafA expression levels among strains were attributable to variations in the putative promoter region, we compared the sequences upstream of the *bafA* gene across 13 strains. The upstream region of the *bafA* gene was identical within each BafA variant type (data not shown). Comparisons of BafA1 and BafA2 strains revealed that the sequences of the −10 box region were completely identical (Fig. S3). However, the −35 box region differed by a single nucleotide. Additionally, a 4 bp (TTTT) deletion in the region 26 bp upstream of the start codon was observed in the *bafA*2 gene.

To rule out the effects of adhesion factors, each strain was cocultured without direct contact with HUVECs using Transwell culture inserts, a method used in our previous study (Fig. 3B) (18). The results obtained showed the significant proliferation of HUVECs over that in the medium control for strains Houston-1, HJ53, PL18, Oki.cat48, and i6-1, ranging from a minimum of 1.16-fold (HJ53 and i6-1) to a maximum of 1.33-fold (PL18) (Fig. 3C). On the other hand, HUVECs cocultured with strains 87–66, IZM2, IZM3, IZM4, IZM5, 5762A, HJ58, and HJ90 did not show significant differences from that in the medium control. The amount of BafA secreted in the supernatants obtained from these cocultures was compared by western blotting. A band of BafA1 was observed at around 51 kDa in the four strains (Houston-1, 87–66, HJ53, and PL18) carrying *bafA1* (Fig. 3D). The amount of BafA in the culture supernatants varied among these strains, with relative band intensities of 4.51 for Houston-1, 1.0 for 87–66, 2.47 for HJ53, and 6.28 for PL18.

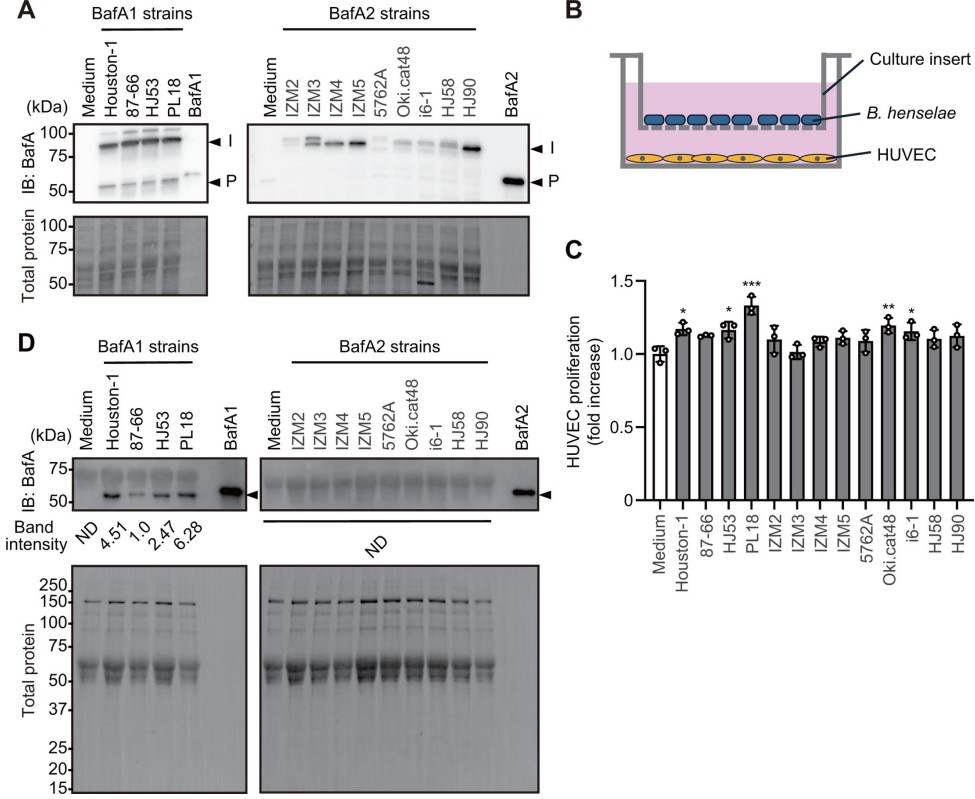

**FIG 3** BafA expression and secretion in the *B. henselae*-HUVEC coculture. (A) Western blot detection of BafA in bacterial cell lysates from HUVEC infection (top) and total protein visualization (bottom). Arrowheads: intact BafA (I) and a cleaved passenger domain (P). Recombinant BafA1 and BafA2 (2 ng each) confirm anti-BafA antibody reactivity. (B) Schematic of the indirect *B. henselae*-HUVEC coculture using Transwell culture inserts. (C) Relative HUVEC proliferation after the 3-day indirect coculture with *B. henselae* strains. Bars: means ± SDs (*n* = 3 biological replicates; open circles). A one-way ANOVA with Dunnett's test: *, $P < 0.05$; **, $P < 0.01$; ***, $P < 0.001$. (D) Western blot detection of BafA in the indirect coculture supernatant (top) and total protein visualization (bottom). Arrowheads: BafA. Relative band intensity shown between images (ND, not detectable).

On the other hand, the BafA band was not observed at around 51 kDa in the nine strains carrying *bafA2* (IZM2, IZM3, IZM4, IZM5, 5762A, Oki.cat48, i6-1, HJ58, and HJ90). Cytotoxicity, as shown in Fig. 1, was not observed in this experiment, although the amount of bacteria equivalent to a MOI of 1,000 was added. These results suggest that Houston-1, 87–66, HJ53, and PL18 have the ability to promote the proliferation of HUVECs when cocultured even without direct contact between the bacterial cells and HUVECs. Variations in this ability may be attributed to the amount of BafA secreted into the culture supernatant, and BafA1 strains have been suggested to differ in their proteolytic processing of BafA. On the other hand, BafA2 strains differed in the amount of BafA expressed within bacterial cells, and are also considered to be incapable of both the proteolytic processing and secretion of BafA.

## BadA expression and bacterial surface structure

The levels of proliferation of HUVECs differed between the direct coculture (Fig. 1) and indirect coculture (Fig. 3C) of *B. henselae* strains and HUVECs, suggesting the involvement of adhesion between bacterial cells and HUVECs in this process. BadA is a TAA on the bacterial surface (15) and a well-established adhesion factor for bacterial cells to adhere to host cells. BadA is a high-molecular-weight protein, with monomers ranging between 296 and 464 kDa, due to the highly repetitive neck/stalk region. The number of repeats varies depending on the strain, resulting in corresponding variations in molecular weight (31). We compared BadA expression levels across strains using western blotting. We observed multiple high-intensity bands, including trimers, monomers, and their degradation products (15, 32, 33) in strains IZM2, IZM3, IZM4, IZM5, 5762A, Oki.cat48, i6-1, HJ58, and HJ90, which carry *bafA2*. Conversely, comparable bands were not detected in strains carrying *bafA1* (Houston-1, 87–66, HJ53, and PL18) (Fig. 4A). This result suggests that BafA2 strains express BadA at high levels. We then examined the cell surface of each strain using scanning electron microscopy. BafA1 strains exhibited smooth surfaces devoid of filamentous structures, which are typically associated with TAA (Fig. 4B). On the other hand, filamentous structures causing cell aggregation were observed in BafA2 strains. These results suggest that strains carrying *bafA2* highly expressed BadA, whereas those carrying *bafA1* did not.

## Phylogenetic relationship of *B. henselae* strains based on whole genome sequences

To investigate the genetic distances between *B. henselae* strains and their relationship with specific *bafA* gene variants, we performed whole genome sequencing of the *B. henselae* strains tested, except for Houston-1, the genome data of which are already publicly available. In the analysis, we included, in addition to the 13 *B. henselae* strains, 37 of 55 *B. henselae* strains registered in the NCBI database strains whose genome data contained a full-length *bafA* gene. These 50 *B. henselae* strains were divided into four well-supported clusters or clades, including two subclades, by a core genome SNP (cgSNP) analysis (Fig. 5A). Among the 13 *B. henselae* strains, BafA1 strains (Houston-1, 87–66, HJ53, and PL18) and BafA2 strains (IZM2, IZM3, IZM4, IZM5, 5762A, Oki.cat48, i6-1, HJ58, and HJ90) each formed a clade, and the former was designated Clade 1, and the latter, Clade 2 a. A comparison of the amino acid sequences of the BafA passenger domain of these 50 strains revealed that strains in Clades 3 and 4 had BafA variants 3 (BafA3) and variant 4 (BafA4) with sequences that differed from those of BafA1 and BafA2. None of the 13 *B. henselae* strains used in our experiments belonged to Clade 3 or 4. The amino acid sequences of the passenger domain were identical among the strains within each clade for both BafA3 and BafA4 variants. A comparative analysis of the putative amino acid sequences of each full-length BafA variant revealed 69% to 76% identity across comparisons (Fig. 5B). Furthermore, passenger domains exhibited 53% to 66% identity (Fig. S4). MLST of 50 *B. henselae* strains classified them into ST1 (21/50 strains), ST5 (17/50 strains), ST6 (5/50 strains), ST7 (2/50 strains), ST9 (3/50 strains), and ST11 (2/50 strains). Twelve of 21 ST1 strains were assigned to Clade 1, and the remaining

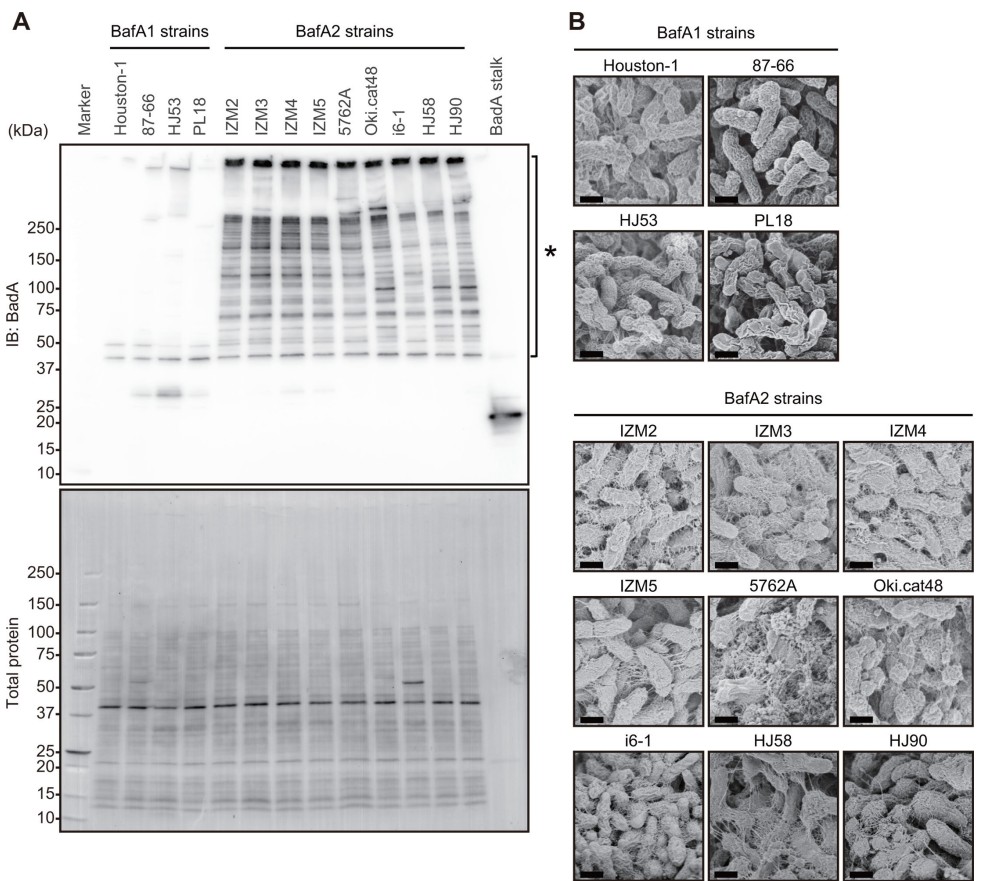

**FIG 4** BadA expression and bacterial surface structures. (A) Western blot of BadA from *B. henselae* strains (top) and total protein visualization (bottom). The recombinant BadA stalk fragment (20 ng/lane) confirms antibody reactivity. Asterisk: multiple BadA-derived bands. (B) Scanning electron microscopy of bacterial surfaces. Strains Houston-1, 87–66, HJ53, and PL18 show smooth surfaces. Strains IZM2-5, 5762A, Oki.cat48, i6-1, HJ58, and HJ90 display filamentous structures and aggregation. Scale bar = 0.5 µm.

9 to Clade 2 a. All three ST9 strains belonged to Clade 1, all 17 ST5 strains to Clade 2b, all five ST6 strains and both ST11 strains to Clade 3, and both ST7 strains to Clade 4. Of the cat-derived strains, 2/26 were in Clade 1, 2/26 in Clade 2 a, 15/26 in Clade 2b, 5/26 in Clade 3, and 2/26 in Clade 4. Of the human-derived strains, 7/15 were in Clade 1, 5/15 in Clade 2 a, 2/15 in Clade 2b, and 1/15 in Clade 3. Of the mongoose-derived strains, 1/3 was in Clade 1 and 2/3 in Clade 2 a. One masked palm civet-derived strain was in Clade 1. These results suggest a close relationship between the four clades defined by the cgSNP analysis of 50 *B. henselae* strains and the BafA variants they produced.

## DISCUSSION

*B. henselae* is the most common causative agent of human bartonellosis, the pathogenesis of which among immunocompromised individuals primarily stems from vasoproliferation (6). We previously identified the angiogenic factor BafA, which is secreted by the bacterium and binds to VEGF receptor-2 in host cells to activate vascular endothelial growth signals (18). In addition to *B. henselae*, other causative agents of human bartonellosis include *B. quintana* and *B. bacilliformis*, which are responsible for trench fever and Carrion's disease, respectively (34, 35). Additional *Bartonella* species, such as *B. koehlerae*, *B. clarridgeiae*, *B. grahamii*, and *B. elizabethae*, have recently been associated with human infection (36). Our previous findings showed that all the *Bartonella* species causing diseases in humans possessed the *bafA* gene and also that BafA secreted by each species exerted distinct proliferation-promoting effects on HUVECs (18–21). BadA,

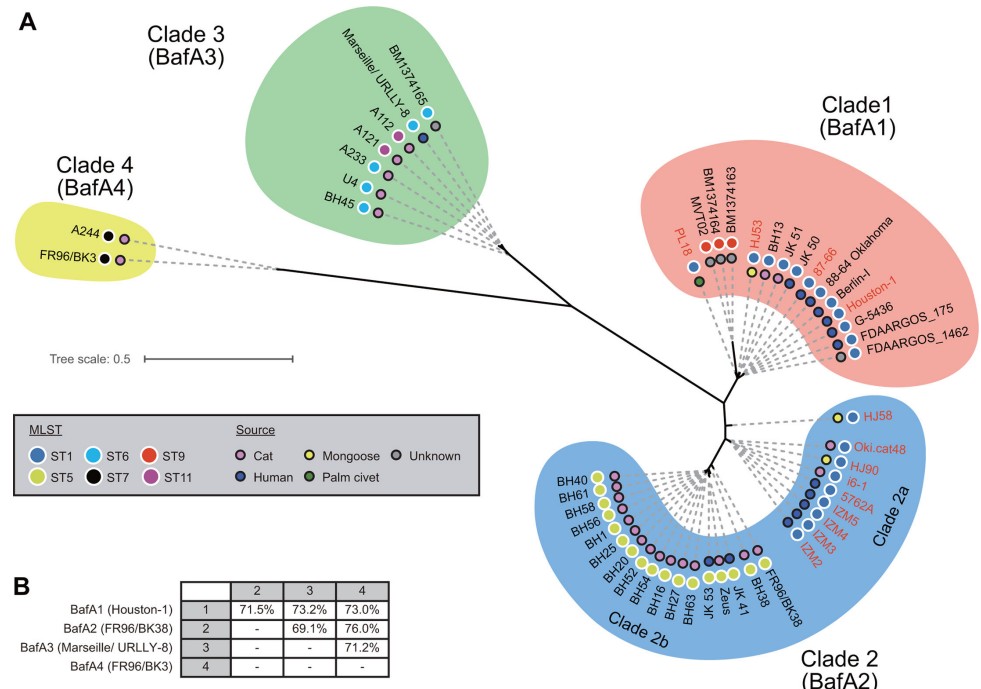

**FIG 5** Phylogenetic analysis of *B. henselae* strains based on whole genome sequences. (A) Core genome SNP-based phylogenetic tree. Clades correspond to BafA variants. Red: strains used in this study. Gray box: Sequence type (ST) from multilocus sequence typing (MLST) and the isolate source. (B) BafA variant amino acid sequence comparison. The upper comparison shows the percentage of identical residues in overlapping alignment positions.

a well-characterized factor produced by *B. henselae*, plays a pivotal role in two critical processes: adhesion to host cells and the induction of angiogenic reprogramming through the activation of hypoxia-inducible factor (HIF)−1 (15, 29, 33, 37). The subsequent activation of HIF-1 in host cells stimulates the expression of various vasculoproliferative cytokines, including VEGF, IL-8, and adrenomedullin (29). BadA acts by adhering to host cells (38), is required for biofilm formation composed of *B. henselae* (17, 39), and may play a critical role in the development of endocarditis (2). However, it was unclear whether proliferation-promoting abilities against HUVECs differed among *B. henselae* strains. In the present study, we revealed diversity in the proliferation-promoting abilities of *B. henselae* strains against HUVECs and demonstrated its relationship with the specific BafA variants produced. Furthermore, we showed that some *B. henselae* strains exhibited cytotoxicity against HUVECs at a higher bacterial inoculum.

In the direct coculture of *B. henselae* strains with HUVECs, the degree of proliferation-promoting ability and cytotoxicity for HUVECs varied by each strain, allowing the strains to be divided into two groups based on the observed phenotypes. Strains possessing BafA1 (Houston-1, 87–66, HJ53, and PL18) exhibited significant proliferation-promoting ability and negligible cytotoxicity against HUVECs. Strain 87–66 did not exert significant HUVEC proliferation-promoting effects at a MOI of 100 or 1,000 but weakly induced the proliferation of HUVECs in both direct and indirect cocultures. Moreover, strains possessing BafA2 (IZM2, IZM3, IZM4, IZM5, 5762A, Oki.cat48, i6-1, HJ58, and HJ90) exhibited cytotoxicity at a high bacterial inoculum. These strains were further divided into two groups based on their effects at a low inoculum: those that exhibit no HUVEC proliferation-promoting ability (IZM2, IZM3, IZM4, IZM5, and 5762A) and those that exert significant proliferative effects (Oki.cat48, i6-1, HJ58, and HJ90). Strains Oki.cat48 and i6-1 exhibited significant HUVEC proliferation-promoting ability in both direct and indirect cocultures, distinguishing them from BafA2 strains. However, BafA was not detected in the culture supernatant of the indirect coculture with these bacterial cells and HUVECs.

This result suggests that in these strains, BafA is secreted in amounts that are below the detection threshold of western blotting or that other unidentified secreted factors may be involved in the observed proliferative effect. The modest cell proliferation observed in our HUVEC proliferation assay, ranging from 1.09-fold to 1.29-fold, may be attributed to the experimental conditions and the state of the cells used. To maintain cell viability and ensure accurate counting, we limited the incubation time to 48 h because prolonged exposure to BafA2 strains may have resulted in excessive HUVEC mortality. This shorter incubation period may have resulted in a lower observed proliferation rate. Notably, under identical conditions, even the VEGF treatment resulted in only a 1.22-fold increase in proliferation (Fig. S2). Despite these seemingly low rates, we contend that they represent physiologically meaningful values within the context of the present study.

On the other hand, a common characteristic observed in BafA2 strains is their cytotoxicity, as evidenced by a decrease in the number of HUVECs as the bacterial inoculation increased, suggesting the direct involvement of some cytotoxic effect between the bacteria and HUVECs. Therefore, we examined the mRNA expression of four cytokines in HUVECs during infection to investigate potential factors responsible for the differences in cell proliferative capacity and cytotoxicity among bacterial strains. However, the results obtained did not conclusively explain these differences. Consistent with previous studies (18, 29, 30), only the expression of IL-8 increased during infection across strains. Notably, the cytotoxic strains IZM5 and 5762A also induced significant increases in IL-8 mRNA levels, despite their detrimental effects on HUVECs. This suggests the involvement of an unidentified cytotoxic factor that overrides the vasoproliferative effects typically associated with IL-8. In the study by Chang et al. (22), human microvascular endothelial cell line-1 was infected with four strains of *B. henselae* (BafA2 strains were not included) at MOIs of 50 and 100 proliferated, and no cytotoxicity was observed. These findings, in combination with the present results, suggest that *B. henselae* not only promotes the proliferation of vascular endothelial cells but also exhibits cytotoxicity; however, the extent of these abilities varies depending on the strain, and the cytotoxicity of some strains becomes more pronounced at a higher inoculum.

In the indirect coculture experiment, the amount of BafA secreted differed between BafA1 strains. Moreover, strains with higher levels of BafA detected in the culture supernatant (e.g., PL18) exerted more significant HUVEC proliferation-promoting effects. This result indicates a relationship between the amount of BafA secreted into the culture supernatant and proliferation-promoting effects on HUVECs. However, taken together with the results shown in Fig. 2C, the amount of BafA is not necessarily proportional to the proliferation-promoting activity against HUVECs. These results suggest that variations in cell proliferation-promoting activities and cytotoxicity among *B. henselae* strains are not solely due to the secretion of BafA. They may be attributed to multiple factors, including the secretion of BafA, the infection rate, the ability to enter cells, intracellular survival, and host modulation.

BafA expression was consistent and similar across all BafA1 strains but was negligible or absent in some BafA2 strains. This difference between BafA1 and BafA2 strains may be attributed to a few nucleotide substitutions or deletions in the putative promoter region. However, within each variant type, the promoter regions are identical, suggesting the involvement of unidentified activator(s) or repressor(s) in the regulation of BafA expression. Notably, in addition to the intact size of BafA, a band of approximately 50 kDa, likely corresponding to the passenger domain, was detected in BafA1 strains, but not in any BafA2 strains. The BafA autotransporter is considered to undergo cleavage, releasing a ~50 kDa passenger domain. Therefore, this processing may be impaired in BafA2 strains, potentially explaining why BafA secretion was not detected in these strains.

The present results revealed no obvious BadA expression in four BafA1 strains and high expression in nine BafA2 strains. Thibau et al. recently reported that active recombination within the repetitive genomic *badA* island facilitated the reshuffling of homologous domain arrays, leading to variations in BadA expression (31). In Houston-1

strains, ATCC49882$^T$ var-1 did not express BadA, whereas ATCC49882$^T$ var-2 showed high expression, which suggested that our Houston-1 strain corresponds to ATCC49882$^T$ var-1. Thibau et al. also detected the expression of BadA in BafA1 strains 88–64 Oklahoma and G-5436. Based on these findings, no clear relationship exists between the phylogenetic clade and BadA expression. The inverse relationship between BafA secretion and BadA expression in the present study may be coincidental. However, we cannot exclude the potential involvement of BadA in the secretion of BafA. To elucidate this relationship, further analyses of additional strains, including BadA-expressing BafA1 strains (e.g., ATCC49882$^T$ var-2), are needed.

In summary, we demonstrated that *B. henselae* exhibited both proliferation-promoting ability and cytotoxicity toward vascular endothelial cells, with the degree of these abilities being dependent on the BafA variant types they possess. In other words, our results indicate that *B. henselae* exerts these contradictory effects on endothelial cells and that their balance leads to differences in the strain phenotype. Furthermore, the clades separated by cgSNPs correlated well with the *bafA* gene variants possessed by each clade. These results suggest a relationship between the genomic background of the strains and the BafA variant types. In the present study, only 13 of the 50 strains used in the cgSNP analysis were physiologically available. To elucidate the potential relationship between phylogenetic clades and the vasoproliferative characteristics of their respective strains, it is important to expand the sample size of BafA1 strains and analyze additional strains, including those from Clades 3 and 4. This expanded analysis will provide a more comprehensive understanding of the relationships across different clades. Based on the present results, we speculate that cell proliferation-promoting strains may cause bacillary angiomatosis, whereas cytotoxic strains are prone to cause conditions such as endocarditis. Collectively, the results obtained herein provide important novel insights into the pathogenicity of *B. henselae*.

## MATERIALS AND METHODS

### Bacterial strains and cell culture

The 13 *B. henselae* strains used in the present study are listed in Table 1. *B. henselae* strains Houston-1 (ATCC 49882) and 87–66 (ATCC 49793) were purchased from the American Type Culture Collection (Manassas, VA). Strains IZM2, IZM3, IZM4, and IZM5 from a human; i6-1 and Oki.cat48 from cats; HJ53, HJ58, and HJ90 from mongooses; and PL18 from a masked palm civet were isolated in previous studies (40–42). Strain 5762A was collected from a CSD patient at Fujita Health University Hospital in Aichi, Japan. Bacteria were grown on Columbia agar with 5% defibrinated sheep blood (Becton Dickinson, Franklin Lakes, NJ) in a 5% $CO_2$ humidified atmosphere at 37°C. HUVECs

**TABLE 1** Bacterial strains used in the present study

| Strain | Source | Country |
|---|---|---|
| Houston-1 (ATCC 49882) | Human (HIV-positive, fever) | USA: Houston |
| 87–66 (ATCC 49793) | Human (AIDS) | USA: Oklahoma |
| IZM2 | Human (cat-scratch disease) | Japan: Nagasaki |
| IZM3 | Human (cat-scratch disease) | Japan: Nagasaki |
| IZM4 | Human (cat-scratch disease) | Japan: Nagasaki |
| IZM5 | Human (cat-scratch disease) | Japan: Nagasaki |
| 5762A | Human (cat-scratch disease) | Japan: Aichi |
| Oki.cat48 | Cat | Japan: Okinawa |
| i6-1 | Cat | Japan: Gunma |
| HJ53 | Small Indian mongoose | Japan: Okinawa |
| HJ58 | Small Indian mongoose | Japan: Okinawa |
| HJ90 | Small Indian mongoose | Japan: Okinawa |
| PL18 | Masked palm civet | Japan: Chiba |

acquired from PromoCell (Heidelberg, Germany) were cultivated in endothelial growth medium 2 (EGM-2; PromoCell) in a 5% $CO_2$ humidified atmosphere at 37°C. Experiments were conducted between passages 5 and 7.

## Antibodies for western blotting

A synthetic peptide of BafA2 (amino acid residues 110–128: EEKSLRNLDSLDSINKRSS) and the recombinant BadA stalk fragment (amino acid residues 377–539) were used as antigens for antiserum production. The peptide was synthesized by Eurofins Genomics (Tokyo, Japan), which also performed the immunization of Japanese white rabbits. The purification of each antibody from antisera was conducted using rProtein A Sepharose Fast Flow (Cytiva, Marlborough, MA). The polyclonal antibody against BafA1 was prepared in a previous study (18). The specificities of the BafA antibodies are shown in Fig. S5. The anti-BafA1 antibody specifically recognized BafA1. In contrast, the anti-BafA2 antibody not only bound to BafA2 but also exhibited some non-specific reactions with other proteins.

## Detection of the *bafA* gene

To establish whether *B. henselae* strains carried the *bafA* gene, we performed colony-direct PCR on 13 *B. henselae* strains cultured for 7 days. One colony was selected from each strain and suspended in 20 µL of sterilized water, and the bacterial suspension was then heated at 95°C for 3 min. The complete *bafA* gene, including its putative promoter region predicted by the BPROM program (Softberry, Inc., Mount Kisco, NY), was amplified using Platinum SuperFi DNA polymerase (Thermo Fisher Scientific, Waltham, MA) with the primers BafA-305-Fw and BafA-93-Rv. Heat extracts from *B. henselae* colonies were used as templates. The PCR protocol consisted of an initial denaturation at 98°C for 30 s, followed by 35 cycles at 98°C for 10 s, 56°C for 15 s, and 72°C for 90 s. The reaction concluded with a final extension at 72°C for 5 min. The primers were synthesized by Thermo Fisher Scientific. PCR products were purified using the MinElute PCR Purification Kit (Qiagen, Hilden, Germany) and then sequenced by Sanger sequencing at Eurofins Genomics. Regarding strains Houston-1, 87–66, HJ53, and PL18, the following primers were used for sequencing: BafA-305-Fw, BafA-93-Rv, BafA1-Seq-Fw, and BafA12-Seq-Rv. The primers used for the other nine strains were as follows: BafA-305-Fw, BafA-93-Rv, BafA2-Seq-Fw, and BafA12-Seq-Rv (Table S1).

## Plasmid construction

The plasmid vector for the expression of the passenger domain of BafA1 (amino acids 25–497) was previously generated from *B. henselae* Houston-1 (18). To prepare the passenger domain of BafA2, the plasmid pET-28b-BAF2 was generated as follows. Nucleotides encoding the 25th to 497th amino acids of BafA were amplified with the primer set NheI-BAF2-Fw—NheI-BAF2-Rv using the full-length *bafA* gene derived from *B. henselae* HJ90 as the template. PCR fragments were purified by the MinElute PCR Purification Kit and inserted into NheI-digested pET-28b (Sigma-Aldrich, St. Louis, MO) using the In-Fusion HD cloning kit (TaKaRa Bio Inc., Shiga, Japan). To prepare the plasmid expressing the BadA stalk fragment, *badA* encoding amino acid residues 377–539 (stalk region) was amplified with the primer set NheI-BadA-Fw – NheI-BadA-Rv using genomic DNA from strain Houston-1 as the template. The insertion of PCR products into the pET-28b vector was conducted in the same manner as described above. All sequences inserted into the plasmid were verified using Sanger sequencing (Eurofins Genomics).

## Expression and purification of recombinant proteins

Recombinant BafA1 (amino acid residues 25–497) was expressed and purified in a previous study (18). Recombinant BafA2 and the BadA stalk fragment were expressed and purified in the same manner as BafA1. Briefly, *Escherichia coli* BL21 (DE3) harboring

expression plasmids were cultivated in LB medium with 25 µg/mL kanamycin. Protein expression was induced by adding isopropyl-β-D-thiogalactopyranoside (20 µM) at 14°C overnight. The lysis of harvested *E. coli* cells was performed on ice by sonication in 25 mM Tris-HCl (pH 7.5), 500 mM NaCl, and 30 mM imidazole with a protease inhibitor cocktail (Nacalai Tesque, Kyoto, Japan). The purification of recombinant proteins from the lysate was conducted through two chromatography steps, Ni-Sepharose affinity chromatography (HisTrap HP column; Cytiva) and gel filtration (Superdex 200 column; Cytiva). Protein concentrations were measured using the Protein Assay BCA Kit (FujiFilm Wako, Osaka, Japan).

## HUVEC proliferation by the direct coculture with *B. henselae*

Following a 5-day cultivation of the bacterial strain in the manner described above, bacteria were harvested from the culture plate and suspended in Medium 199 (M199; Thermo Fisher Scientific) supplemented with 5% fetal bovine serum (FBS; Biowest, France). Measurements of the optical density at 600 nm ($OD_{600}$) of the bacterial suspension showed that $OD_{600}$ of 1.0 corresponded to $1 \times 10^9$ bacterial cells/mL. HUVECs (7,000 cells/well) were seeded onto a gelatin-coated 96-well plate with EGM-2. After HUVECs were incubated for 4 h, the medium was replaced with M199/5% FBS. Each bacterial suspension was added to the HUVEC culture with MOI = 100, 300, or 1,000 and incubated for 2 days. Cells were stained with Hoechst 33342 (1 µg/mL; Nacalai Tesque) and ViaFluor 488 SE Cell Proliferation Dye (0.5 µM; Biotium, Fremont, CA) at 37°C for 5 min. Hoechst 33342 is a nuclear stain, whereas ViaFluor 488 SE selectively labels mammalian cells without staining Gram-negative bacteria. After staining, the solution was replaced with M199/5% FBS, and the plate was incubated at 37°C for 5 min. The fixation of cells was performed with 4% paraformaldehyde at room temperature for 15 min and was followed by rinsing with phosphate-buffered saline (PBS). Images of the stained cells were acquired using the CellVoyager CQ1 confocal quantitative image cytometer (Yokogawa Electric Corporation, Tokyo, Japan). In each well, 10 fields of view were captured at ×10 magnification, and the average cell number was calculated from seven wells per experimental group using high-content analysis software, Cell Pathfinder (Yokogawa Electric Corporation).

## HUVEC proliferation by recombinant BafA

HUVECs (7,000 cells/well) were seeded onto a 96-well plate with EGM-2. Cells were incubated, and the medium was exchanged as described under "HUVEC proliferation by the direct coculture with *B. henselae*." Cells were then treated with the indicated concentrations of recombinant BafA1 or BafA2 and cultured for 2 days. The proliferation of HUVECs was evaluated using the method previously outlined.

## Indirect coculture of HUVEC and *B. henselae*

To coculture HUVECs with *B. henselae* without direct contact, Transwell Polycarbonate Membrane Insert (6.5 mm, 0.4 µm pore; Corning, Corning, NY) was used to separate them. HUVECs (45,000 cells/well) were plated onto a gelatin-coated 24-well plate with EGM-2. After HUVECs were incubated for 5 h, the medium was replaced with 600 µL of M199/2% FBS, and Transwell inserts were inserted into the plate. The bacterial suspension of each strain was collected and adjusted with $OD_{600}$ of 0.45 ($4.5 \times 10^8$ bacterial cells/mL) as described above. After 100 µL of the bacterial suspension was added to the Transwell inserts, the plate was incubated for 3 days. CellVoyager CQ1 was used to assess the proliferation of HUVECs as described above.

## Detection of BafA from cell lysates and culture supernatants

To investigate BafA expression in bacteria infecting HUVECs, we cultured each *B. henselae* strain with HUVEC in 24-well plates (45,000 cells/well) at an MOI of 1,000 for 2 days.

After the incubation, we removed the medium and added 50 µL of BugBuster Master Mix (Merck Millipore, Burlington, MA) to each well in order to lyse both bacterial cells and HUVECs. The collected lysate was mixed with 10 µL of Sample Buffer Solution containing Reducing Reagent (6×) for SDS-PAGE (Nacalai Tesque) in preparation for subsequent SDS-PAGE and western blotting analyses. To examine the secretion of BafA from each *B. henselae* strain, bacteria were indirectly cocultured with HUVECs as described above. Three hundred microliters of the culture supernatant were centrifuged to remove bacterial cells. To deplete albumin, 300 µL of Minute albumin depletion reagent for plasma and serum (Invent Biotechnologies, Inc., Plymouth, MN) was added. The supernatant (albumin fraction) was removed after recentrifugation. Cells were then washed with 600 µL of ultrapure water, and the precipitate was resuspended in 30 µL of PBS. Six microliters of Sample Buffer Solution were added to the residual sample and heated at 95°C for 3 min. Lysates and culture supernatant samples both underwent SDS-PAGE using a 10% Mini-PROTEAN TGX Stain-Free Protein Gel (Bio-Rad, Hercules, CA) in Running Buffer Solution for SDS-PAGE (Nacalai Tesque). After electrophoresis was completed, total protein fluorescence signals in the gel were excited by 45 s of UV irradiation using a ChemiDoc Touch Imaging System (Bio-Rad). Proteins were then transferred onto a polyvinylidene difluoride (PVDF) membrane using the Trans-Blot Turbo RTA Mini PVDF Transfer Kit (Bio-Rad). After washing the membrane with TBS, images of total proteins were obtained by the ChemiDoc Touch Imaging System. The iBind solution kit (Thermo Fisher Scientific) was used for blocking and antibody incubation following the manufacturer's instructions. The primary antibodies (anti-BafA1, 5 µg/mL; anti-BafA2, 10 µg/mL) and secondary antibody (HRP-Donkey anti-rabbit IgG, 1:4,000, Jackson ImmunoResearch, West Grove, PA) were sequentially probed with the iBind Western System. After the membrane was washed and immersed with Chemi-Lumi One Ultra (Nacalai Tesque) for chemiluminescence, bands were detected using the ChemiDoc Touch Imaging System.

## Scanning electron microscopy

After 7 days of the bacterial strain culture in the manner described above, bacterial cells were collected and suspended in heart infusion broth. After centrifugation and removal of the supernatant, bacterial pellets were mounted on a membrane filter (pore size of 0.22 µm) and fixed with 1% formaldehyde and 1.25% glutaraldehyde in 0.05 M cacodylate buffer on ice for 15 min. Specimens were subsequently washed and postfixed using 1% osmium tetroxide in 0.05 M cacodylate buffer with 1% potassium ferrocyanide. After dehydration in a graded series of ethanol, specimens were dried using the critical point drying method using the Samdri-PVT-3D system (Tousimis, Rockville, MD). The osmium coater HPC-30W (Vacuum Device, Ibaraki, Japan) was used to coat the specimens with osmium tetroxide. Electron micrographs were obtained with the S-4800 field emission scanning electron microscope (Hitachi, Tokyo, Japan).

## Whole genome sequencing

After culturing the bacterial strain as described above for 7 days, bacterial cells were collected from the culture plate and suspended in PBS. After centrifugation of the suspension, the extraction and purification of genomic DNA were accomplished with the PureLink genomic DNA mini kit (Thermo Fisher Scientific). The DNA library for whole genome sequencing was prepared using the QIAseq FX DNA Library Kit, and sequencing was performed on the Illumina NextSeq 2000 platform using P2 Reagents (300 Cycles) v3 (Illumina, San Diego, CA). Quality trimming of the sequence reads was conducted using fastp (https://github.com/OpenGene/fastp), and the genome assembly was constructed using the SPAdes Genome Assembler, available on GIGAdoc (https://github.com/suzukimasahiro/gigadoc).

## Bioinformatics

Genomes were annotated using DFAST v1.2.0. CLC Main Workbench 8.1.2 software (Qiagen) was used to construct the alignment of multiple amino sequences. BadA nucleotide and amino acid sequences were acquired from the *B. henselae* Houston-1 genome sequence available at NCBI (accession no. BX897699). MLST was performed on the PubMLST website (https://pubmlst.org/). The cgSNP analysis was conducted using SNIPPY, available on GIGAdoc as described above, with Houston-1 (accession no. BX897699) as the reference. The phylogenetic analysis based on cgSNPs was performed using FastTree (http://www.microbesonline.org/fasttree), and the unrooted phylogenetic tree was visualized using iTOL v6. Table S2 lists the NCBI accession numbers of the sequences used in this study.

## Statistical analysis

Data were expressed as means with standard deviations (SD). To compare multiple groups, a one-way ANOVA with Dunnett's multiple-comparison test was performed as indicated in the figure legends. These analyses were conducted using GraphPad Prism 10 software (GraphPad Software, La Jolla, CA). Each experiment was repeated at least three times to confirm the reproducibility of the results obtained. Representative western blots and microscope photographs were selected from a minimum of three independent biological replicates, which all yielded similar outcomes.

### ACKNOWLEDGMENTS

We thank Hiroko Omori for her technical assistance with the ultrastructural analysis, and Yusuke Minato and Akito Kawai for their helpful comments. This research was supported in part by the Japan Society for the Promotion of Science (JSPS) KAKENHI, JP22K070602 (to K.T.); the Japan Agency for Medical Research and Development (AMED) Japan Program for Infectious Diseases Research and Infrastructure under grant number JP23wm0325058 (to K.T.); a Grant for the Joint Research Project of the Research Institute for Microbial Diseases, Osaka University (to S.S.). The efforts of Y.D. were supported in part by the National Institutes of Health (R01AI014895).

Y.K. performed most of the laboratory experiments, analyzed data, and drafted the manuscript. M.S. and A.S. contributed to whole genome sequencing and the cgSNP analysis. S.S. and S.M. isolated and cultured *B. henselae* strains. X.M. supported the cocultivation of bacteria with HUVECs. K.N. contributed to the phylogenetic analysis. Y.D. supervised the research and revised the manuscript. K.T. designed and performed the research, analyzed data, and revised the manuscript.

### AUTHOR AFFILIATIONS

[1]Laboratory of Bacterial Zoonoses, International Research Center for Infectious Diseases, Research Institute for Microbial Diseases, Osaka University, Suita, Osaka, Japan

[2]Department of Microbiology, Fujita Health University School of Medicine, Toyoake, Aichi, Japan

[3]Department of Veterinary Medicine, College of Bioresource Sciences, Nihon University, Fujisawa, Kanagawa, Japan

[4]Department of Infectious Diseases, Fujita Health University School of Medicine, Toyoake, Aichi, Japan

[5]Division of Infectious Diseases, University of Pittsburgh School of Medicine, Pittsburgh, Pennsylvania, USA

### AUTHOR ORCIDs

Yuka Kondo (iD) http://orcid.org/0009-0006-9419-3190
Masahiro Suzuki (iD) http://orcid.org/0000-0003-4550-3499
Shingo Sato (iD) http://orcid.org/0000-0002-6837-977X

Kentaro Tsukamoto 🆔 http://orcid.org/0000-0002-9254-4632

## FUNDING

| Funder | Grant(s) | Author(s) |
|---|---|---|
| MEXT \| Japan Society for the Promotion of Science (JSPS) | JP22K070602 | Kentaro Tsukamoto |
| Japan Agency for Medical Research and Development (AMED) | JP23wm0325058 | Kentaro Tsukamoto |
| OU \| Research Institute for Microbial Diseases, Osaka University (RIMD) | | Kentaro Tsukamoto |
| HHS \| National Institutes of Health (NIH) | R01AI014895 | Yohei Doi |

## AUTHOR CONTRIBUTIONS

Yuka Kondo, Formal analysis, Investigation, Visualization, Writing – original draft | Masahiro Suzuki, Investigation, Software | Shingo Sato, Investigation, Resources | Soichi Maruyama, Investigation, Resources | Akiko Sei, Investigation | Xingyan Ma, Investigation | Kota Nakano, Formal analysis | Yohei Doi, Writing – review and editing | Kentaro Tsukamoto, Conceptualization, Data curation, Funding acquisition, Investigation, Methodology, Project administration, Resources, Supervision, Validation, Writing – review and editing

## DATA AVAILABILITY

The genomes of *B. henselae* strains were deposited in the DDBJ Sequence Read Archive under DDBJ BioProject number PRJDB17373. Upon reasonable request, raw Sanger sequencing data may be obtained from the corresponding author. This article includes all other data generated in this study.

## ADDITIONAL FILES

The following material is available online.

### Supplemental Material

**Supplemental figures (Spectrum01925-24-s0001.pdf).** Fig. S1 to S5.
**Supplemental material (Spectrum01925-24-s0002.docx).** Supplemental methods.
**Supplemental tables (Spectrum01925-24-s0003.docx).** Tables S1 and S2.

### Open Peer Review

**PEER REVIEW HISTORY (review-history.pdf).** An accounting of the reviewer comments and feedback.

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
