## [Reviewer comments · Microbiology Spectrum]

Microbiology Spectrum

Differential vasoproliferative traits of *Bartonella henselae* strains associated with autotransporter BafA variants

Yuka Kondo, Masahiro Suzuki, Shingo Sato, Soichi Maruyama, Akiko Sei, Xingyan Ma, Kota Nakano, Yohei Doi, and Kentaro Tsukamoto

Corresponding Author(s): Kentaro Tsukamoto, Osaka Daigaku Biseibutsubyo Kenkyujo

Review Timeline:

Submission Date:	August 5, 2024
Editorial Decision:	August 29, 2024
Revision Received:	October 25, 2024
Accepted:	November 1, 2024

Editor: John Attack

Reviewer(s): The reviewers have opted to remain anonymous.

Transaction Report:

DOI: <https://doi.org/10.1128/spectrum.01925-24>

Re: Spectrum01925-24 (Strain-dependent characteristics of *Bartonella henselae* in vascular endothelial cell proliferation)

Dear Dr. Kentaro Tsukamoto:

Thank you for the privilege of reviewing your work. Below you will find my comments, instructions from the Spectrum editorial office, and the reviewer comments.

Revision Guidelines

Sincerely,
John Attack
Editor
Microbiology Spectrum

Reviewer #1 (Comments for the Author):

In this manuscript, the authors describe the differences in expression of the autotransporter BafA and a possible link to endothelial cell proliferation. For this, the authors included 13 *B. henselae* strains in their experiments. They found two isoforms of BafA (BafA1, BafA2), expressed these proteins recombinantly and analysed the influence of these proteins on EC proliferation. Additionally, they compared the BafA expression with the expression of BadA in these isolates. Finally they

present a phylogenetic analysis based on whole genome analysis, and defined four clades based on four different BafA variants.

Major remarks:

Language and grammar need to be improved throughout the manuscript (e.g., by a native speaker).

Line 120-121 and 163 and 170: My most relevant point of criticism is the relatively low biological effect of EC proliferation which is only between 1.09 to 1.28 fold. In my opinion such an effect is very low (although statistically significant). I cannot judge whether this effect detected in in vitro assays might have any biological effect in vivo. Author must clearly explain this in much more detail.

Principally: I do not understand why the authors have not quantified at least some of the cytokines which have been described earlier to be induced by *B. henselae*. This is easy and might help to understand the results better (especially the relatively low induction of EC proliferation)

Principally: Authors should include an experiment in which they confirm the expression of BafA1 or BafA2 by the respective strains (e.g., immunofluorescence, Western blotting).

Line 132 ff: Authors describe that recombinant BafA1 and BafA2 induce EC proliferation (Fig. 2). I essentially miss a negative control (e.g., a different recombinant protein of *B. henselae*)

Line 190: It has been shown several times that *B. henselae* Houston I strains differ in BadA expression (e.g., Thibau et al., Front Microbiol 2022). So, therefore, the authors seem to work with the so-called var-1 strain. Therefore, generalization of their results is definitively not possible, as they use a strain variant in BadA expression. Therefore, the conclusions made in the discussion (line 289 ff) are not valid and must be corrected.

Line 350 ff: Specificity of the BafA antibodies must be confirmed and described.

Minor remarks:

Title: the title is unspecific and should be much more concrete.

Line 32: virB/D TIVSS should be included here.

Line 40: "Mature BadA": it is not clear what "mature" means here.

Line 49: vasculoproliferative lesions are not "part of bacterial pathogenesis" but are the result of the infection process with human pathogenic *B. henselae*. Please correct.

Line 71/72: For my knowledge it has never been shown that bacterial invasion of *B. henselae* into ECs causes EC proliferation. Coculture-based EC proliferation and induction of EC-proliferation by vasculoproliferative cytokines have been described.

Line 186: BadA is not a "pilus-like structure" but a so called TAA (or OCA). Please correct.

Line 191: BadA is not a "fimbrial structure" but a so called TAA (or OCA). Please correct.

Line 243: vasculoproliferative cytokines: VEGF and other factors have been identified, please complete and cite. Moreover, hypoxic gene programming has been described several times (e.g., Kempf et al., Circulation 2005). Please correct.

Spectrum01925-24: Strain-dependent characteristics of *Bartonella henselae* in vascular endothelial cell proliferation

General comment: This is a very interesting paper! In this study, authors investigated proliferation-promoting activity of 13 *B. henselae* strains derived from several sources (humans, cats, mongooses, and masked palm civet) in human umbilical vein endothelial cells (HUVECs). Several species of pathogenic gram-negative bacterial genus *Bartonella*, including *B. henselae*, the most common causative agent of human bartonellosis, can promote vascular proliferation and induce vasoproliferative lesions in animals and humans. The mechanisms underlying *Bartonella*-triggered vasoproliferation remains unclear. BadA and BafA, two *Bartonella* autotransporters, have been identified as key virulence factors involved in pathogenicity of *Bartonella species* including promoting endothelial cell proliferation. Findings from this study provide molecular and cellular insights into the proliferation-promoting ability and pathogenicity of *B. henselae* strains. The authors noted that expression profile of BafA and BadA, and the degree of proliferation-promoting ability in vascular endothelial cells varied among the *B. henselae* strains based on variant types.

Overall, the manuscript is well written, and the research findings are scientifically sound. However, in my opinion, the manuscript would benefit from additional editing and experiments. Despite many strengths of the present study, “Results” and “Discussion” sections require some clarification. Results of this study about i) *B. henselae* proliferation-promoting and cytotoxicity activities towards HUVECs (Fig. 1) and ii) expression levels of BafA and BadA in each *B. henselae* strain were interpreted based on the experiments involving four *B. henselae* BafA variant 1 strains (n=4) and nine *B. henselae* BafA variant 2 strains (n=9). Inclusion of a relatively large sample size for *B. henselae* BafA variant 1 strains would be relevant and helpful for understanding the association between the BafA variants and effects on vascular endothelial cells. Do the authors want to comment on how this (inclusion of more *B. henselae* BafA variant 1 strains) affect their interpretation of results? I appreciate the authors attempt to perform core genome SNPs analysis among 50 *B. henselae* strains. Based on the deduced amino acid sequences of the BafA passenger domains, authors noted that BafA variants exhibited 53% to 66% identity across comparisons. Recombinant BafA variants 3 and 4 and *B. henselae* strains with BafA variants 3 and 4 belonging to clades 3 and 4, respectively, were not tested for their proliferation-promoting ability in vascular endothelial cells in this study. Given that none of the 13 *B. henselae* strains tested in this study had BafA variants 3 and 4, I think assessment of proliferation-promotion effects of BafA variants 3 and 4 on vascular endothelial cells further strengthen findings of this study.

These specific comments will hopefully be of benefit to the authors and readers of this manuscript. Once these questions/comments are addressed, I would recommend this manuscript for publication.

Specific comments:

Line 39: On line 39, please consider adding “at MOI of 1000” or “at high bacterial inoculum” after “...cytotoxicity...” as “... cytotoxicity at MOI of 1000...”. BafA variant 2 strains exhibited proliferation activity at MOIs of 100 and 300 and cytotoxicity at MOI of 1000.

Line 59-60: ...”suggest clade-specific pathogenicity...”. Please consider revising this statement. The BafA variant types corresponded to clades 1 to 4 based on core genome SNPs analysis. The effects of

BafA variants 3 and 4 strains on HUVECs were not experimentally tested and compared in this study. I think experiments conducted in this study may not prove the clade-specific pathogenicity of *B. henselae* strains.

Line 66-67: "...acquired through direct contact....via flea infestation." Please add a reference.

Line 104: "... BafA secretion and BadA expression, differ depending on the clades.....". Please consider revising this statement. The author discussed on lines 305 to 310, "association between differences in BafA variants and BadA expression was not always consistent". Per my earlier comment, BafA secretion and BadA expression of *B. henselae* strains corresponding to clade 3 and 4 was not tested in this study. In Fig. 3 (C), BafA expression varied among *B. henselae* variant 1 strains (strains belonging to clade 1). It might be better to have more experimental data to suggest that BafA secretion and BadA expression differ depending on clades. It would be helpful to comment on the fact that clade-specific- pathogenicity (on line 59), BafA secretion, and BadA expression. Please clarify.

Line 112 to128: As author stated, proliferation-promoting activity varied among *B. henselae* strains. The only strain that showed significant HUVEC proliferation effect at MOI of 100, 300, and 1000 was PL18 (BafA variant 1, Fig. 1), the strain with higher levels of BafA secretion (relative band intensity 6.28, Fig. 3 (C)) in the culture supernatant. In contrast to this observation, proliferating activity of recombinant BafA variant 1 against HUVECs was not significantly different for each concentration. Perhaps, variability in proliferation-promotion activity and cytotoxicity among *B. henselae* strains can be attributed to several factors, for example, differences in infectivity rate, ability to enter the cells, intracellular survival, and host modulation. These factors should be acknowledged/addressed in the results or discussion sections.

Also, authors PCR-amplified the complete *bafA* gene including the putative promoter region in this study (line 362-363). Can the BafA expression variability be explained by the differences in *bafA* promoter sequences?

Line 157 to179: BafA band was not detected in the culture supernatant among nine BafA variant 2 strains when cocultured without direct contact with HUVECs. What experimental approach authors took to confirm the expression of BafA variant 2 among these BafA variant 2 strains when cocultured with HUVECs directly or indirectly? For example, RNA extraction and RT-PCR or ELISA testing can be performed. Please clarify.

Line 187-189: "...multiple bands derived from BadA.....". Please clarify what these multiple bands represent? Do these bands represent degradation products of BadA?

Line 236: " and Oroya fever, respectively". Please consider replacing "Oroya fever" with "Carrion disease". Please add a reference.

Line 250: "...cytotoxicity against HUVECs". To avoid any confusion to the readers, please consider adding "at higher bacterial inoculum" after HUVECs.

Line 324: Please consider adding "In this study," before "BafA variant 2 strains express...".

Line 351: Please provide amino acids residues for recombinant BafA Variant 2 that was used for antiserum production. Is it nucleotides encoding 25th to 497th amino acid as stated on line 374?

Line 363: Which tool was used to determine the putative promoter region of *bafA* gene? Please include the PCR conditions/steps used for amplification of *bafA* gene. Were the PCR amplicons confirmed by sequencing? Please clarify this in the methods sections.

Line 371: It might be better to provide amino acid residue information for the passenger domain of BafA variant 1 in this study as well.

Line 393: Please provide the amino acid residue information for recombinant BafA variant 1 in this study as “Recombinant BafA variant 1 (amino acids XX-XX) was expressed.....”.

Line 414: Were the HUVECs cocultured with *B. henselae* (after 2 days incubation as indicated on line 414) treated with gentamicin to kill extracellular and membrane-bound bacteria before staining? Please clarify.

Line 699: Please indicate how much recombinant BadA stalk fragment was loaded as control for BadA western blotting?

Responses to Reviewers' & Editor's Comments

We sincerely appreciate the prompt response from the editor and reviewers, as well as the opportunity to revise our manuscript. Your insightful comments and suggestions have been invaluable, guiding us to conduct additional experiments and deepen our analyses. In response, we have mainly:

1. Examined cytokine mRNA expression in HUVEC (Fig. S1)
2. Added positive and negative controls to Fig. 2C (Fig. S2)
3. Compared the bafA gene's putative promoter region between BafA variant strains (Fig. S3)
4. Confirmed anti-BafA antibody specificity in western blotting (Fig. S5)
5. Analyzed BafA expression in bacterial cells (Fig. 3A)
6. Summarized primer sequences in Table S1 and Text S1 for supplemental methods
7. Significantly revised the Discussion section to address reviewers' points
8. Adopted "BafA1" and "BafA2" terminology for concision throughout the manuscript

These additions and revisions have substantially strengthened our manuscript. We are pleased to resubmit our revised work, now entitled "Differential vasoproliferative traits of *Bartonella henselae* strains associated with autotransporter BafA variants". We believe these changes address all raised concerns and hope that our revised manuscript meets the standards for publication in *Microbiology Spectrum*. We extend our sincere gratitude to the editor and reviewers for their time, effort and valuable input.

*The comments provided by the editor and reviewers are copied in blue italics. Our responses are shown in black Roman type and the revised position in red (figure, page and line numbers).

Reviewer comments:

Reviewer #1 (Comments for the Author):

*In this manuscript, the authors describe the differences in expression of the autotransporter BafA and a possible link to endothelial cell proliferation. For this, the authors included 13 *B. henselae* strains in their experiments. They found two isoforms of BafA (BafA1, BafA2), expressed these proteins recombinantly and analyzed the influence of these proteins on EC proliferation. Additionally, they compared the BafA expression with the expression of BadA in these isolates. Finally they present a phylogenetic analysis*

based on whole genome analysis, and defined four clades based on four different BafA variants.

Thank you for reviewing our manuscript.

Major remarks:

Language and grammar need to be improved throughout the manuscript (e.g., by a native speaker).

We appreciate your feedback regarding the language and grammar in our manuscript. We apologize for any errors and have taken your suggestion seriously. To improve the quality of our writing, we engaged a native English-speaking editor with expertise in scientific editing to thoroughly review and correct the manuscript. A certificate of proofreading is attached at the end of this document for your reference.

1. Line 120-121 and 163 and 170: My most relevant point of criticism is the relatively low biological effect of EC proliferation which is only between 1.09 to 1.28 fold. In my opinion such an effect is very low (although statistically significant). I cannot judge whether this effect detected in in vitro assays might have any biological effect in vivo. Author must clearly explain this in much more detail.

We appreciate your valuable suggestion. In response to the reviewer's feedback, we have revised several sentences to address the issue of redundant notation of detailed proliferation rate numbers, which previously obscured the clear explanation of our results (lines 116-125, 196-199). The observed modest cell proliferation, ranging from 1.09 to 1.29-fold, may be attributed to the experimental condition and the state of the cells used. To maintain cell viability and ensure accurate cell counting, we limited the incubation time to 48 hours, as prolonged exposure to the BafA2 strains would have resulted in excessive HUVEC mortality. This shorter incubation period likely contributed to the lower observed proliferation rate. Importantly, under identical experimental conditions, even VEGF treatment resulted in only a 1.22-fold increase in proliferation (Fig. S2). Therefore, we contend that these seemingly low proliferation rates still represent physiologically meaningful values within the context of our study. I have added this explanation to Discussion (lines 314-324).

2. Principally: I do not understand why the authors have not quantified at least some of the cytokines which have been described earlier to be induced by B. henselae. This is easy and might help to understand the results better (especially the relatively low induction of EC proliferation)

We appreciate your advice. In response, we conducted further experiments to examine the mRNA expression of four cytokines: VEGF, IL-8, TNF- α , and IL-6. Although the results did not conclusively explain the differences in proliferation-promoting ability and cytotoxicity observed among the strains in the infection experiment, they provide valuable insights. We have included the results of these additional experiments in **Fig. S1**. A description of the results and related discussion have been added to lines **129-142** and **329-337** of the manuscript, with the detailed method provided in **Text S1**. It is important to note that cytokine mRNA expression does not always directly correlate with protein levels due to post-transcriptional regulation and other factors. Our next study will focus on elucidating the specific molecular mechanisms of cytotoxicity exhibited by particular strains. Given this future direction, we kindly request your understanding in limiting our current discussion of the molecular mechanisms to the level presented in this paper. We believe this approach allows us to present our current findings while acknowledging the need for further investigation, which we are committed to pursuing.

3. Principally: Authors should include an experiment in which they confirm the expression of BafA1 or BafA2 by the respective strains (e.g., immunofluorescence, Western blotting).

Thank you for highlighting this point. We have conducted additional experiments to address your concern. After infecting HUVECs with *B. henselae*, we collected the bacterial cells with HUVECs and detected BafA from their combined lysate. This approach confirms the expression of BafA during HUVEC infection. We have incorporated the results from this additional experiment into Fig. 3, now presented as **Fig. 3A**. The related results, discussion, and methodological description have been added to lines **178-184**, **358-368**, and **518-524** of the manuscript.

4. Line 132 ff: Authors describe that recombinant BafA1 and BafA2 induce EC proliferation (Fig. 2). I essentially miss a negative control (e.g., a different recombinant protein of B. henselae)

In accordance with the reviewer's suggestion, we have conducted additional control experiments. Specifically, we used recombinant BadA stalk as a negative control and VEGF as a positive control in our HUVEC proliferation assays. The results of these experiments are presented in **Fig. S2**, and related explanations were added to lines **163-171** of the text.

5. Line 190: *It has been shown several times that B. henselae Houston I strains differ in BadA expression (e.g., Thibau et al., Front Microbiol 2022). So, therefore, the authors seem to work with the so-called var-I strain. Therefore, generalization of their results is definitively not possible, as they use a strain variant in BadA expression. Therefore, the conclusions made in the discussion (line 289 ff) are not valid and must be corrected.*

Thank you for your valuable comments. As the reviewer noted, the conclusion regarding the association between BadA expression and BafA variant strains was not appropriate, and we have revised the relevant paragraph in Discussion accordingly (lines 370-382).

6. Line 350 ff: *Specificity of the BafA antibodies must be confirmed and described.*

In response to the reviewer's suggestion, we have addressed the specificity of the BafA antibodies. We have included a demonstration of their reaction specificity in Fig. S5, and added a relevant description to lines 423-426.

Minor remarks:

1. Title: *the title is unspecific and should be much more concrete.*

The title has been changed to "Differential vasoproliferative traits of *Bartonella henselae* strains associated with autotransporter BafA variants".

2. Line 32: *virB/D TIVSS should be included here.*

As suggested by the reviewer, we have added the phrase "VirB/D4 type IV secretion system". (line 32).

3. Line 40: *"Mature BadA": it is not clear what "mature" means here.*

The word "mature" is unclear, as the reviewer pointed out, so we rephrase it as "...highly expressed BadA..."(line 40).

4. Line 49: *vasculoproliferative lesions are not "part of bacterial pathogenesis" but are the result of the infection process with human pathogenic B. henselae". Please correct.*

Following the reviewer's comments, and taking into account the 150 word limit, we have removed the incorrect phrase (lines 49-50).

5. Line 71/72: *For my knowledge it has never been shown that bacterial invasion of B. henselae into ECs causes EC proliferation. Coculture-based EC proliferation and induction of EC-proliferation by vasculoproliferative cytokines have been described.*

We have revised the sentence as suggested (lines 71-74).

6. Line 186: *BadA is not a "pilus-like structure" but a so called TAA (or OCA). Please correct.*

The wording was replaced as suggested (line 222).

7. Line 191: *BadA is not a "fimbrial structure" but a so called TAA (or OCA). Please correct.*

In response to the reviewer's remarks and to more accurately reflect our electron microscopic observations, we have revised the term "fimbrial structures" to "filamentous structures, which are typically associated with TAA" in this description (lines 233-234).

8. Line 243: *vasculoproliferative cytokines: VEGF and other factors have been identified, please complete and cite. Moreover, hypoxic gene programming has been described several times (e.g., Kempf et al., Circulation 2005). Please correct.*

We have revised the sentences as you suggested and added the relevant references (lines 283-288).

Reviewer #2 (Comments for the Author):

General comment: This is a very interesting paper! In this study, authors investigated proliferation-promoting activity of 13 B. henselae strains derived from several sources (humans, cats, mongooses, and masked palm civet) in human umbilical vein endothelial cells (HUVECs). Several species of pathogenic gram-negative bacterial genus Bartonella, including B. henselae, the most common causative agent of human bartonellosis, can promote vascular proliferation and induce vasoproliferative lesions in animals and humans. The mechanisms underlying Bartonella -triggered vasoproliferation remains unclear. BadA and BafA, two Bartonella autotransporters, have been identified as key virulence factors involved in pathogenicity of Bartonella species including promoting endothelial cell proliferation. Findings from this study provide molecular and cellular insights into the proliferation-promoting ability and pathogenicity of B. henselae strains. The authors noted that expression profile of BafA and BadA, and the degree of proliferation-promoting ability in vascular endothelial cells varied among the B. henselae strains based on variant types.

Thank you for your positive feedback on our manuscript.

Overall, the manuscript is well written, and the research findings are scientifically sound. However, in my opinion, the manuscript would benefit from additional editing and experiments. Despite many strengths of the present study, “Results” and “Discussion” sections require some clarification. Results of this study about i) B. henselae proliferation-promoting and cytotoxicity activities towards HUVECs (Fig. 1) and ii) expression levels of BafA and BadA in each B. henselae strain were interpreted based on the experiments involving four B. henselae BafA variant 1 strains (n=4) and nine B. henselae BafA variant 2 strains (n=9). Inclusion of a relatively large sample size for B. henselae BafA variant 1 strains would be relevant and helpful for understanding the association between the BafA variants and effects on vascular endothelial cells. Do the authors want to comment on how this (inclusion of more B. henselae BafA variant 1 strains) affect their interpretation of results? I appreciate the authors attempt to perform core genome SNPs analysis among 50 B. henselae strains. Based on the deduced amino acid sequences of the BafA passenger domains, authors noted that BafA variants exhibited 53% to 66% identity across comparisons. Recombinant BafA variants 3 and 4 and B. henselae strains with BafA variants 3 and 4 belonging to clades 3 and 4, respectively, were not tested for their proliferation-promoting ability in vascular endothelial cells in this study. Given that none of the 13 B. henselae strains tested in this study had BafA variants 3 and 4, I think assessment of proliferation-promotion effects of BafA variants 3 and 4 on vascular endothelial cells further strengthen findings of this study.

These specific comments will hopefully be of benefit to the authors and readers of this manuscript. Once these questions/comments are addressed, I would recommend this manuscript for publication.

Thank you for your insightful remarks. We acknowledge the limitations you have highlighted in our study, particularly the insufficient number of BafA variant 1 strains tested and the absence of BafA variant 3 and 4 strains. a larger sample size and a broader range of strains would be valuable for elucidating the relationship between BafA variants and their vasoproliferative characteristics. This point has also been incorporated into the Discussion (lines 391-397).

Specific comments:

1. Line 39: On line 39, please consider adding “at MOI of 1000” or “at high bacterial inoculum” after “....cytotoxicity...” as “... cytotoxicity at MOI of 1000...”. BafA variant 2 strains exhibited proliferation activity at MOIs of 100 and 300 and cytotoxicity at MOI of 1000.

We have added the phrase “at high bacterial inoculum” as suggested (lines 38-39).

2. Line 59-60: *... ”suggest clade-specific pathogenicity... ”. Please consider revising this statement. The BafA variant types corresponded to clades 1 to 4 based on core genome SNPs analysis. The effects of BafA variants 3 and 4 strains on HUVECs were not experimentally tested and compared in this study. I think experiments conducted in this study may not prove the clade-specific pathogenicity of B. henselae strains.*

Following the reviewer’s suggestion, we have replaced the term “clade-specific” with “differential vasoproliferative potency” (lines 58-59). In addition, we have changed the term “... differ depending the clades to which the strain belong” to “... differed among the strains tested,...” (line 105).

3. Line 66-67: *“...acquired through direct contact....via flea infestation.”. Please add a reference.*

We have added references 3-5 (line 68).

4. Line 104: *“... BafA secretion and BadA expression, differ depending on the clades..... ”. Please consider revising this statement. The author discussed on lines 305 to 310, “association between differences in BafA variants and BadA expression was not always consistent”. Per my earlier comment, BafA secretion and BadA expression of B. henselae strains corresponding to clade 3 and 4 was not tested in this study. In Fig. 3 (C), BafA expression varied among B. henselae variant 1 strains (strains belonging to clade 1). It might be better to have more experimental data to suggest that BafA secretion and BadA expression differ depending on clades. It would be helpful to comment on the fact that clade-specific-pathogenicity (on line 59), BafA secretion, and BadA expression. Please clarify.*

We acknowledge that our limited experimental data (small sample size for Clade 1 and lack of experimental data for Clade 3 or 4 strains) make the term 'Clade-specific' an overstatement. Consequently, we have revised our wording and description (lines 58-59, 105, 391-397).

5. Line 112 to128: *As author stated, proliferation-promoting activity varied among B. henselae strains. The only strain that showed significant HUVEC proliferation effect at MOI of 100, 300, and 1000 was PL18 (BafA variant 1, Fig. 1), the strain with higher levels of BafA secretion (relative band intensity 6.28, Fig. 3 (C)) in the culture supernatant. In contrast to this observation, proliferating activity of recombinant BafA*

variant 1 against HUVECs was not significantly different for each concentration. Perhaps, variability in proliferation-promotion activity and cytotoxicity among B. henselae strains can be attributed to several factors, for example, differences in infectivity rate, ability to enter the cells, intracellular survival, and host modulation. These factors should be acknowledged/addressed in the results or discussion sections.

Thank you for your suggestion. We have included this point in our discussion (lines 350-356).

Also, authors PCR-amplified the complete bafA gene including the putative promoter region in this study (line 362-363). Can the BafA expression variability be explained by the differences in bafA promoter sequences?

Thank you for your question. We have provided a comparison of the putative promoter region sequences between BafA variants 1 and 2 in Fig. S3, and added a relevant description in lines 184-192 and 359-363.

6. Line 157 to 179: BafA band was not detected in the culture supernatant among nine BafA variant 2 strains when cocultured without direct contact with HUVECs. What experimental approach authors took to confirm the expression of BafA variant 2 among these BafA variant 2 strains when cocultured with HUVECs directly or indirectly? For example, RNA extraction and RT-PCR or ELISA testing can be performed. Please clarify.

This point, also raised by reviewer 1, prompted us to investigate BafA expression in bacteria during direct coculture with HUVECs using western blotting. The results are now presented in Fig. 3A, with a related description added to lines 178-184, 358-368, and 518-524.

7. Line 187-189: “....multiple bands derived from BadA.....”. Please clarify what these multiple bands represent? Do these bands represent degradation products of BadA?

We have added the explanation you suggested, along with the relevant reference (lines 223-226).

8. Line 236: “ and Oroya fever, respectively”. Please consider replacing “Oroya fever” with “Carrion disease”. Please add a reference.

The wording was replaced, and the reference was added as suggested (line 278).

9. Line 250: “....cytotoxicity against HUVECs”. To avoid any confusion to the readers, please consider adding “at higher bacterial inoculum” after HUVECs.

The words were added as suggested (line 295).

10. Line 324: Please consider adding “In this study,” before “BafA variant 2 strains express...”.

The mentioned section has been removed during the discussion revision (line 388).

11. Line 351: Please provide amino acids residues for recombinant BafA Variant 2 that was used for antiserum production. Is it nucleotides encoding 25th to 497th amino acid as stated on line 374?

A synthetic peptide was used for the production of antiserum against BafA variant 2. We apologize for the error in the initial manuscript and have included the corrected information (line 417-421).

12. Line 363: Which tool was used to determine the putative promoter region of bafA gene? Please include the PCR conditions/steps used for amplification of bafA gene. Were the PCR amplicons confirmed by sequencing? Please clarify this in the methods sections.

We have added the description as suggested (lines 433-445).

13. Line 371: It might be better to provide amino acid residue information for the passenger domain of BafA variant 1 in this study as well.

We have added the information about the amino acid residue (lines 448-449).

14. Line 393: Please provide the amino acid residue information for recombinant BafA variant 1 in this study as “Recombinant BAfA variant 1 (amino acids XX-XX) was expressed.....”.

We have added the information about the amino acid residue (line 464).

15. Line 414: Were the HUVECs cocultured with *B. henselae* (after 2 days incubation as indicated on line 414) treated with gentamicin to kill extracellular and membrane-bound bacteria before staining? Please clarify.

We appreciate your inquiry. To clarify, gentamicin treatment was not performed prior to staining. This is because ViaFluor 488 SE Cell Proliferation Dye specifically stains mammalian cells and does not stain gram-negative bacteria. Consequently, it does not interfere with HUVEC cell count. We have added an explanation of this point (lines 488-489).

16. Line 699: Please indicate how much recombinant BadA stalk fragment was loaded as control for BadA western blotting?

We have added the loading amount of recombinant BadA stalk fragment (line 790).

Re: Spectrum01925-24R1 (Differential vasoproliferative traits of *Bartonella henselae* strains associated with autotransporter BafA variants)

Dear Dr. Kentaro Tsukamoto:

Your manuscript has been accepted, and I am forwarding it to the ASM production staff for publication. Your paper will first be checked to make sure all elements meet the technical requirements. ASM staff will contact you if anything needs to be revised before copyediting and production can begin. Otherwise, you will be notified when your proofs are ready to be viewed.

Sincerely,
John Attack
Editor
Microbiology Spectrum